calcium; channel; endomembrane; homeostasis; plant.

**Corresponding author**:
Colin Brownlee.
Email: cbr@mba.ac.uk

**Associate Editor:** Ingo Dreyer

# Cellular calcium homeostasis and regulation of its dynamic perturbation

Colin Brownlee[1,2] and Glen L. Wheeler[1]

[1]Marine Biological Association, The Laboratory, Citadel Hill, Plymouth, UK; [2]School of Ocean and Earth Sciences, University of Southampton, Southampton, UK

## Abstract

Calcium ions ($Ca^{2+}$) play pivotal roles in a host of cellular signalling processes. The requirement to maintain resting cytosolic $Ca^{2+}$ levels in the 100–200 nM range provides a baseline for dynamic excursions from resting levels that determine the nature of many physiological responses to external stimuli and developmental processes. This review provides an overview of the key components of the $Ca^{2+}$ homeostatic machinery, including known channel-mediated $Ca^{2+}$ entry pathways along with transporters that act to shape the cytosolic $Ca^{2+}$ signature. The relative roles of the vacuole and endoplasmic reticulum as sources or sinks for cytosolic $Ca^{2+}$ are considered, highlighting significant gaps in our understanding. The components contributing to mitochondrial, chloroplast and nuclear $Ca^{2+}$ homeostasis and organellar $Ca^{2+}$ signals are also considered. Taken together, a complex picture of the cellular $Ca^{2+}$ homeostatic machinery emerges with some clear differences from mechanisms operating in many animal cells.

## 1. Introduction

Calcium ions ($Ca^{2+}$) play multiple physiological and structural roles across prokaryote and eukaryote kingdoms. Eukaryotes maintain very low baseline cytosolic $Ca^{2+}$ concentration $[Ca^{2+}_{cyt}]$ of around 100–200 nM since $Ca^{2+}$ is essentially toxic at concentrations even in the low micromolar range due, at least in part, to the ability to bind inorganic phosphate (Clapham, 2007). A typical plant cell is faced with an approximately 1,000-fold inwardly directed concentration gradient across the plasma membrane (PM), so maintaining very low $[Ca^{2+}_{cyt}]$ requires highly efficient homeostatic mechanisms. In parallel to mechanisms for keeping $[Ca^{2+}_{cyt}]$ low, cells have evolved mechanisms to allow controlled entry of $Ca^{2+}$ into the cytosol, giving rise to tightly regulated $Ca^{2+}_{cyt}$ elevations that can act to relay signals from the cell surface to downstream response elements in the cell interior. In plants, patterns of $Ca^{2+}_{cyt}$ elevations vary considerably in response to different stimuli and include single transient elevations, lasting a few seconds, repeated oscillatory elevations over longer time periods and more prolonged elevations (Edel et al., 2017; Lenzoni et al., 2018). Moreover, $Ca^{2+}_{cyt}$ elevations may occur uniformly across the cell or maybe highly localized to a particular region (Brownlee & Wheeler, 2023). A further key feature of $Ca^{2+}$ is its ability to bind reversibly to, and affect the activity of, a wide range of cellular regulatory proteins (Clapham, 2007; Edel et al., 2017) and different $Ca^{2+}_{cyt}$ elevation patterns represent signatures that can differentially activate a wide range of downstream elements (including calmodulin (CaM), calmodulin-like (CMLs), calcineurin B-like (CBLs), calcium-dependent protein kinases (CDPKs) and calcium/calmodulin kinases (CCamKs)) in stimulus- and cell-specific manners to bring about specific end responses (Demidchick et al., 2018; Edel et al., 2017; Lenzoni et al., 2018).

The generation of specific $Ca^{2+}_{cyt}$ signals involves the coordinate orchestration of channels, which allow passive movement of $Ca^{2+}$ down its electrochemical potential gradient and active transporters that maintain resting $[Ca^{2+}_{cyt}]$ and return $Ca^{2+}$ to baseline levels, shaping the $Ca^{2+}_{cyt}$ signature. This review provides an assessment of the key components of the $Ca^{2+}$ homeostatic and signalling machinery. We consider some of the most significant recent advances and key questions still to be addressed. These include: What determines the set points for $Ca^{2+}$

homeostasis? What are the relative capacities and roles of the different cellular $Ca^{2+}$ buffering compartments? To what extent do these compartments act as releasable $Ca^{2+}$ stores and how does this vary with different signalling processes?

Addressing these questions requires quantitative assessments of the contribution of individual and coordinated cellular compartments in the regulation of $Ca^{2+}_{cyt}$. Until recently, measurement of concentrations and fluxes into and out of cellular compartments was primarily based on isolated organelles or microelectrode measurements of larger compartments, such as vacuoles. Recent years have witnessed a number of revolutionary advances, including improved electrophysiological approaches and the advent of targeted genetically encoded fluorescent $Ca^{2+}$ reporters, that will provide new insights into the roles of different components of the $Ca^{2+}$ homeostatic machinery and how its perturbation is finely controlled.

## 2. Components of the plant $Ca^{2+}$ homeostat

### 2.1. Cytosolic buffers

A number of theoretical and experimental studies have shown that steady-state $Ca^{2+}_{cyt}$ levels in eukaryotic cells are determined primarily by the balance between influx and efflux mechanisms rather than passive cytosolic buffering (Eisner et al., 2023; Rios, 2010). However, cytosolic $Ca^{2+}$ buffers do play an important role in modulating the rates of change of $[Ca^{2+}_{cyt}]$ as well as the amplitude of $Ca^{2+}_{cyt}$ elevations in response to changes in membrane $Ca^{2+}$ fluxes (Eisner et al., 2023). Consider a typical PM $Ca^{2+}$ channel passing around 0.5 pA or ~$10^6$ $Ca^{2+}$ ions/s in a single cuboid plant cell of volume approximately $10^{-14}$ $m^3$ and a typical cytosolic volume of 10% total cellular volume. A simple calculation reveals that in the absence of any cytosolic buffering a single $Ca^{2+}$ channel with an open probability of 0.5 could potentially raise the $[Ca^{2+}_{cyt}]$ at a rate of ~5 $\mu M$ $s^{-1}$. While there are no reliable estimates of $Ca^{2+}$ channel density in the PM of plant or algal cells, single-channel patch clamp studies (e.g. Taylor et al., 1996; White et al., 1999) suggest a conservative estimate of 0.5 channels $\mu m^{-2}$, which would equate to a total potential channel complement in the thousands. Clearly, if all $Ca^{2+}$ channels in the PM open simultaneously, then without buffering or active mechanisms to remove $Ca^{2+}$, $[Ca^{2+}_{cyt}]$ could potentially elevate at a rate of several mM $s^{-1}$ until equilibrium concentrations were achieved across the PM. A similar calculation based on a typical whole-cell plant $Ca^{2+}$ current of around 100 pA gives a similar rate of $[Ca^{2+}_{cyt}]$ increase in the absence of buffering. Since most whole-cell stimulus-induced $Ca^{2+}$ transients do not reach peak values higher than the low $\mu M$, there must exist highly efficient mechanisms for $Ca^{2+}$ buffering or removal. Passive $Ca^{2+}_{cyt}$ buffers include $Ca^{2+}$-binding proteins, polyvalent inorganic and organic anions anionic lipid heads, and carboxyl residues (Demidchick et al., 2018; Eisner et al., 2023; Schwaller, 2010). While the buffering capacity of plant cytosol has not been precisely determined and will vary with cell type, in a typical animal cell, the presence of a range of $Ca^{2+}$ buffers with Kd values slightly higher than resting $[Ca^{2+}_{cyt}]$, suggests that passive buffering would become more effective as $[Ca^{2+}_{cyt}]$ began to rise above resting levels (Schwaller, 2010). The affinities, concentrations, kinetics and mobilities of the $Ca^{2+}$ buffers will subsequently determine the rate of $[Ca^{2+}_{cyt}]$ elevation and its amplitude (Eisner et al., 2023; Neher, 1998; Wagner & Keizer, 1994), as well as the extent of $Ca^{2+}_{cyt}$ gradients resulting from localized PM $Ca^{2+}$ fluxes in polarized

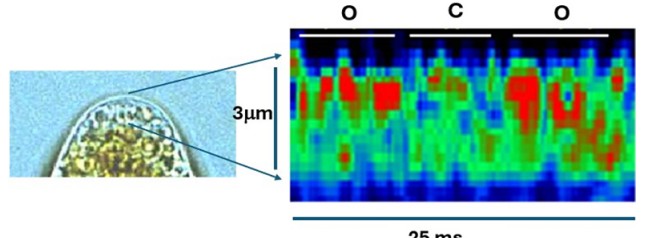

**Figure 1.** Highly localized $Ca^{2+}$ elevations, visualized with the fluorescent $Ca^{2+}$ indicator Calcium Green dextran, at the apex of a *Fucus serratus* rhizoid cell during the initiation of a $Ca^{2+}_{cyt}$ elevation in response to hypoosmotic treatment. Discrete $Ca^{2+}$ elevations (red) are apparent and do not propagate >1 μm from the PM during putative channel opening (O) and disappear during channel closure (C). Adapted from Goddard et al. (2000).

plant cells, such as pollen tubes (e.g. Pierson et al., 1994). The ratio of bound/free $Ca^{2+}$ in plant cytosol has been estimated to be >90% (Demidchick et al., 2018; Schonknecht & Bethmann, 1998) and as little as 1% of total $Ca^{2+}_{cyt}$ is considered to be free in a typical animal cell (Eisner et al., 2023), implying that an elevation of free $[Ca^{2+}_{cyt}]$ from 200 nM to 2 μM would require an influx sufficient to increase total $Ca^{2+}_{cyt}$ by >20 μM. The impact of strong cytosolic $Ca^{2+}$ buffering is also evident from a wide range of animal studies that have shown highly localized $Ca^{2+}$ elevations (sparks) at sites of $Ca^{2+}$ entry through channels that only propagate further through coordinated $Ca^{2+}$-dependent $Ca^{2+}$ release from intracellular stores (Cheung & Lederer, 2008). Non-equilibrium buffering of $Ca^{2+}$ influx through a single channel is dependent on the rate of $Ca^{2+}$ diffusion from the mouth of the channel and the probability that a $Ca^{2+}$ ion will encounter the $Ca^{2+}$ binding site of a buffer molecule, which is critically dependent on the buffer concentration (Stern, 1992). The extent of $[Ca^{2+}_{cyt}]$ increase at the channel mouth will also depend on the rate of diffusion of $Ca^{2+}$-buffer away from the channel as well as the buffer affinity, resulting in an exponential spatial $[Ca^{2+}_{cyt}]$ decay profile (Stern, 1992). While there are no direct examples of such elemental $Ca^{2+}$ elevations in vascular plants, these have been observed in rhizoid cells of the brown alga *Fucus serratus* in response to osmotically-induced channel activation on the PM and endomembranes (Goddard et al., 2000) (Figure 1).

### 2.2. $Ca^{2+}$ entry pathways

**2.2.1. Apoplast-plasma membrane.** The apoplast represents the primary source of $Ca^{2+}$ entering a plant cell with variable apoplastic $[Ca^{2+}]$ reported from a number of studies (Figure 2; e.g. Felle & Hanstein, 2007; Conn et al., 2011; Stael et al., 2012). This, coupled with the large negative PM membrane potential (Vm) produces a large inward-directed electrochemical potential gradient ($\Delta\mu Ca^{2+}$). $Ca^{2+}$ entry across the plasma membrane occurs primarily via $Ca^{2+}$-permeable channels, including cyclic nucleotide-gated channels (CNGCs), glutamate receptors (GLRs), and mechanosensitive channels (MSLs, MCas and OSCAs) (Basu & Haswell, 2017; Brownlee & Wheeler, 2023; Demidchick et al., 2018; Edel et al., 2017; Guichard et al., 2022; Jiang & Ding, 2023; Tian et al., 2020; Yoshimura et al., 2021). Nucleotide-binding leucine-rich repeat receptors (NLRs) mediate immune responses and cell death in response to pathogens. Two plant NLRs (N REQUIREMENT GENE 1 (NRG1) and ZAR1) have also been shown to form $Ca^{2+}$ channels in *Arabidopsis* involved in $Ca^{2+}$-mediated resistance to pathogen attack (Bi et al., 2021; Jacob et al., 2021).

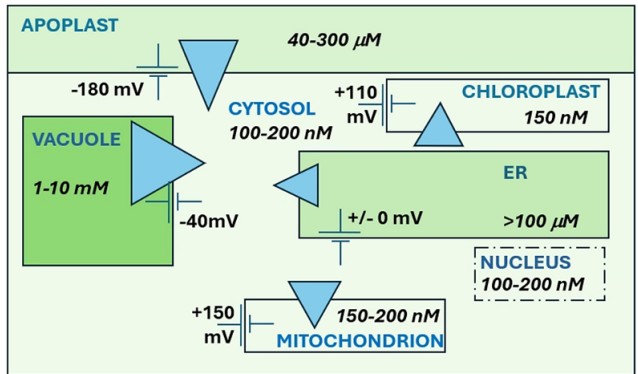

**Figure 2.** $Ca^{2+}$ concentrations, gradients and membrane potentials in a typical plant cell. Blue triangles represent the magnitude and direction of the electrochemical potential gradient ($\Delta\mu Ca^{2+}$).

$Ca^{2+}$ entry channels can be activated in highly specific manners by a very wide range of biotic and abiotic stimuli and can be subject to multiple forms of regulation, including voltage-dependent activation and inactivation (Hille, 1978). For, example, in root hairs, CNGCs, including CNGC5, CNGC6, CNGC9 and CNGC14 play important roles in apically localised $Ca^{2+}$ signalling (Tan et al., 2020; Tian et al., 2020). Notably, CNGC14 is inhibited by elevated $[Ca^{2+}_{cyt}]$ via calmodulin (Zeb et al., 2020). In pollen tubes the $Ca^{2+}$- permeable CNGC18/CNGC8 heterotetramer is preferentially localized at the growing tip and becomes active through interaction with calmodulin (CaM2) at low $[Ca^{2+}_{cyt}]$, leading to increased $Ca^{2+}$ influx (Frietsch et al., 2007; Gao et al., 2014; Pan et al., 2019; Tian et al., 2020). GLRs also contribute significantly to the shaping of the pollen tube $Ca^{2+}_{cyt}$ gradient (Michard et al., 2017; Tian et al., 2020) and regulation of their activity and localization involves CORNICHON homologues (AtCNIHs) (Wudick et al., 2018). In stomatal guard cells, three classes of channels are involved in the closure response to external stimuli and are differentially regulated in response to external cues. The kinase BIK1 activates OSCA1.3 in response to bacterial flagellin (flg22) (Thor et al., 2020). Abscisic acid (ABA) was recently shown to activate CNGC channels via the $Ca^{2+}$-independent kinase OST1 (Yang et al., 2024). GLR channels, activated by external L-methionine were also shown to be involved in stomatal $Ca^{2+}$ signalling, involving further $Ca^{2+}$ channel activation via reactive oxygen (ROS) production (Kong et al., 2016). There are also numerous reports of depolarization- and hyperpolarization-activated $Ca^{2+}$ channels in plants for which detailed electrophysiological information is available (Demidchick et al., 2018). Genes encoding depolarization-activated $Ca^{2+}$ channels have only been identified in animals and those encoding hyperpolarization currents in guard cells or root hairs remain unidentified.

While $Ca^{2+}$ entry across the PM may initiate elevation of $[Ca^{2+}_{cyt}]$ during signalling events, there is good evidence that in many cases $Ca^{2+}$ release from intracellular stores may account for the bulk of $Ca^{2+}$ entering the cytosol:

### 2.2.2. Channel-mediated Vacuolar $Ca^{2+}$ release.
The vacuole may occupy more than 90% of the total cell volume in a typical plant cell. While vacuolar free $[Ca^{2+}]$ ($[Ca^{2+}_{vac}]$) can vary substantially (Pottosin & Schonknecht, 2007), microelectrode measurements indicate $[Ca^{2+}_{vac}]$ in the low mM range (Felle, 1988). This, coupled with a cytosol-negative membrane potential (Dindas et al., 2021) generates a large ($\Delta\mu Ca^{2+}$) directed into the cytosol (Figure 2).

Despite the obvious potential to represent the largest releasable $Ca^{2+}$ store, the role of vacuolar $Ca^{2+}$ release during signalling remains unclear. Direct involvement of the $Ca^{2+}$-permeable slow vacuolar (SV) channel TPC1 (Peiter et al., 2005) as a major pathway for vacuolar $Ca^{2+}$ release into the cytosol (Hedrich & Neher, 1987; Ward & Schroeder, 1994) was questioned following the finding that *Arabidopsis Attpc1* mutants did not show any differences from wild type in stomatal $[Ca^{2+}_{cyt}]$ elevations in response to external ABA or methyl jasmonate (Islam et al., 2010). *Attpc1* mutants were able to close their stomata normally in response to ABA but showed either no response (Peiter et al., 2005) or a reduced response to increased external $Ca^{2+}$ (Islam et al., 2010). Moreover, long-distance salinity- or wounding-induced root-shoot $Ca^{2+}$ waves were respectively significantly slower or abolished in *Attpc1* mutants (Choi et al., 2014; Kiep et al., 2015).

Mechanosensitive tonoplast-localized $Ca^{2+}$-permeable (PIEZO) channels have been shown to be involved in the transduction of mechanical signals in *Arabidopsis* root columnellar cells (Mousavi et al., 2021). Chimaeras of AtPIEZO with mouse mPIEZO generated non-selective mechanosensitive currents in HEK cells. PIEZO channels are therefore potentially involved in mechanosensitive $Ca^{2+}$ release from the vacuole. PIEZO channels were also shown to contribute to $Ca^{2+}_{cyt}$ oscillations in the moss *Physcomitrium* and to be an important factor regulating vacuolar morphology (Radin et al., 2021).

### 2.2.3. Endoplasmic reticulum (ER).
While the Vm across the ER membrane has not been directly measured in plants, in animals ER Vm is clamped to near zero by high $K^+$ conductance and equimolar $[K^+]$ on both sides of the ER membrane (Lam & Galione, 2013). However, an ER free $[Ca^{2+}]$ ($[Ca^{2+}_{ER}]$) >100 μM (Daverkausen-Fischer & Pröls, 2022) establishes a large cytosol−directed $\Delta\mu Ca^2$ (Figure 2). While most studies of $[Ca^{2+}_{ER}]$ changes in response to various stimuli do not report calibrated values in plants, lower affinity variants of GECI reporters (e.g. ER-GCAMP6) have been used successfully to report $[Ca^{2+}_{ER}]$ changes (Resentini et al., 2021a). In animals, the $Ca^{2+}$ release pathway from the ER is very well studied with both Inositol 1,4,5 trisphosphate (InsP$_3$) and ryanodine receptors acting as the major pathways for $Ca^{2+}$ release (Katona et al., 2022; Lam & Galione, 2013). However, while there are reports of InsP$_3$-induced $Ca^{2+}$ release in plant cells (e.g. Manzoor et al., 2012; Muir & Sanders, 1996), vascular plants do not possess canonical InsP$_3$ or ryanodine receptors (Edel et al., 2017; Verret et al., 2010) and there is no characterized mechanism for facilitating $Ca^{2+}$ release directly from the ER into the cytosol in plants. A clear exception, however, comes from work with root nodulation and mycorrhizal symbioses responses (Lam et al., 2024). Distinct from plasma membrane channel-mediated increases in $[Ca^{2+}_{cyt}]$, mycorrhizal (Myc) factors from arbuscular mycorrhizal fungi and nodulation (Nod) factors from rhizobia can give rise to $Ca^{2+}$ elevations localized to the nucleus (Charpentier et al., 2016; Lam et al., 2024; Oldroyd, 2013). In the legumes of *Medicago trunculata* or *Lotus japonicus*, Nod factor perception by LysM-type plasma membrane receptors is conveyed to the ER-derived nuclear envelope through a cytosolic mevalonate pathway (Venkateshwaran et al., 2015). Interaction between CNGC15 and the CASTOR/POLLUX/DMI1 channels on the inner nuclear envelope membrane underlies the release of $Ca^{2+}$ into the nucleoplasm. While DMI1 has been proposed to behave as a $K^+$ channel, more recent evidence suggests that both CNGC15 and DMI1 are $Ca^{2+}$ channels (Kim et al., 2019).

**Table 1.** Examples of identified endomembrane transporters with demonstrated roles in modulation of $Ca^{2+}_{cyt}$ or organellar $Ca^{2+}$.

| Intracellular sources and sinks with demonstrated roles in regulating $Ca^{2+}_{cyt}$ | | | | |
|---|---|---|---|---|
| Compartment membrane | Transporters | $Ca^{2+}_{cyt}$ impact | Physiological process | References |
| Tonoplast | PIIB ATPase PCA1 | ↓ | *Physcomitrium* salinity/osmotic signalling | Qudeimat et al., 2008 |
| | TPC1? | ↑<br>↑ | *Arabidopsis* long-distance signalling.<br>Stomatal closure in response to external $Ca^{2+}$ and ROS. | Cho et al., 2012;<br>Islam et al., 2010;<br>Kiep et al., 2015;<br>Peiter et al., 2005 |
| | AtACA4, ACA11 | ↓ | Arabidopsis response to flg22 and maintenance of resting $Ca^{2+}_{cyt}$. | Frei et al., 2012;<br>Hilleary et al., 2020 |
| | AtPIEZO (AtPZO1)<br><br>PpPIEZO | ↑<br>↑ | Mechanotransduction in root cells.<br>$Ca^{2+}_{cyt}$ oscillations in<br>*Physcomitrium* | Mousavi et al., 2021<br>Radin et al., 2021 |
| | AtCAX2 | ↓ | Flooding and hypoxia $Ca^{2+}_{cyt}$ response modulation | Bakshi et al., 2023 |
| | AtCCX2 | ↕? | Salt and osmotic responses | Corso et al., 2018 |
| ER | AtECA1<br><br>AtACA1,2,7<br><br>NbCA1<br>AtCCX2 | ↓<br><br>↓<br><br>↓ | Asymmetric $Ca^{2+}_{cyt}$ elevation in root bending and hydrotropic response.<br>Arabidopsis response to flg and maintenance of resting $Ca^{2+}_{cyt}$.<br>Responses to cryptogenic.<br><br>Regulation of ER-cytosol $Ca^{2+}$ exchange. | Shkolnik et al., 2018<br>Ishka et al., 2021<br>Corso et al., 2018 |
| Intracellular organelle-specific $Ca^{2+}_{cyt}$ elevations | | | | |
| Compartment membrane | Transporters | Organelle $Ca^{2+}$ impact | Physiological process | References |
| Mitochondria | AtMCU1–3<br><br>AtMICU | ↑<br><br>↓ | A major route for fast mitochondrial $Ca^{2+}$ uptake. Required for jasmonic acid signalling and thigmomorphogenesis<br>Modulates mitochondrial $Ca^{2+}$ uptake by MCU and shapes mitochondrial $Ca^{2+}$ signatures. | Ruberti et al., 2022;<br>Teardo et al., 2017<br>Wagner et al., 2015 |
| Chloroplast | AtMCU<br><br>AtBICAT2<br><br>AtBICAT1 | ↑<br><br>↑<br><br>↓ | Stromal $Ca^{2+}$ accumulation in response to hyperosmotic shock.<br>Imports $Ca^{2+}$ across the chloroplast envelope. Underlies stromal $Ca^{2+}$ increase in response to light–dark transition<br>Transports $Ca^{2+}$ into the thylakoid lumen. Modulates BICAT2-mediated stromal $Ca^{2+}$ increase | Teardo et al., 2019<br>Frank et al., 2019 |
| Nucleus | MtCNGC15/MtDMI1 | ↑ | Nuclear envelope-ER localized channels coordinately mediate nuclear $Ca^{2+}_{cyt}$ oscillations in response to root nodulation factors | Liu et al., 2022<br>Charpentier et al., 2016 |

While molecular evidence for specific pathways underlying ER-mediated $Ca^{2+}$ release into the cytosol is lacking, there is clear physiological evidence for a role for ER $Ca^{2+}$ release in $Ca^{2+}_{cyt}$ signalling. The mechanisms of trap closure by the Venus flytrap involve an initial depolarization of the trap lobe cell PM following mechanical stimulation of trap lobe hair cells. This involves an initial influx of $Ca^{2+}$, most likely through GLR3.6 channels (Scherzer et al., 2022). Based on the effects of the ER $Ca^{2+}$-ATPase inhibitor cyclopiazonic acid (CPA), which increased resting $[Ca^{2+}_{cyt}]$ but decreased the amplitude of the transient $Ca^{2+}$ elevation required for the trap closure response, Scherzer et al. (2022) deduced that an initial $[Ca^{2+}_{cyt}]$ elevation associated with $Ca^{2+}$ influx through GLR channels was augmented substantially by $Ca^{2+}$ release from the ER. A similar role for ER $Ca^{2+}$ release in elevation of $[Ca^{2+}_{cyt}]$ has recently been indicated by Huang et al. (2023), who monitored $Ca^{2+}_{cyt}$ in *Arabidopsis* guard cells expressing a light-activated $H^+$-permeable channelrhodopsin (HcKCR2) to generate $H^+$-induced $[Ca^{2+}_{cyt}]$ elevations. Pharmacological treatment with cyclopiazonic acid (CPA) or low apoplastic $Ca^{2+}$ (EGTA) indicated a role in ER $Ca^{2+}$ release. Moreover, repetitive stimulation led to a stepwise reduction of the $Ca^{2+}_{cyt}$ signals, indicating that ER $Ca^{2+}$ stores

could be depleted if $Ca^{2+}$ release exceeded the recharging ability of ER-localized ATPases (see below). More recently, Huang et al. (2024) used the $Ca^{2+}$-permeable, blue light-activated optogenetic probe channelrhodopsin ChR2-XXM2.0 to elicit pulses of $Ca^{2+}$ influx across the PM of stomatal guard cells. This study showed that $Ca^{2+}$ influx gave rise to $Ca^{2+}_{cyt}$ elevations, which were also associated with incremental closure of the stomata. CPA prevented the ChR-induced transient $Ca^{2+}_{cyt}$ elevation, presumably by inhibiting the loading of $Ca^{2+}_{cyt}$ into ER stores and provided evidence for ER store depletion with repetitive (every 6 min) $Ca^{2+}$ release events.

### 2.3. Transporters maintaining $Ca^{2+}_{cyt}$ homeostasis

A combination of empirical and modelling approaches has provided evidence for the recovery phase of $Ca^{2+}$ signals in determining the nature of downstream signalling responses (Lenzoni et al., 2018). The rate of recovery of $Ca^{2+}_{cyt}$ following elevation, and the set point for resting $[Ca^{2+}_{cyt}]$ is largely determined by the affinity and regulated activity of $Ca^{2+}$ extrusion systems on the PM and endomembranes (Table1). Two main classes of $Ca^{2+}$-ATPases remove $Ca^{2+}$ from the cytosol: Type PIIA (ER-type, ECA)

located mainly, but not restricted to endomembranes, unlike animal ECA ATPases, and Type IIB, autoinhibited ATPases (ACA), which can be located at both PM and endomembranes (vacuole, ER, Golgi) (see Costa et al., 2023 for a recent review). ATPases exchange 1 $Ca^{2+}$ for 1 or 2 $H^+$, therefore utilising both the energy of ATP hydrolysis and the $H^+$ electrochemical potential gradient. *Arabidopsis* AtACA8 comprises 10 transmembrane domains and a cytoplasmic head with nucleotide binding and CaM binding domains. Other regulatory mechanisms include phosphorylation (e.g. via CPK1/16, CIPK9/14, CBL1, CaM, CML36) and differences in ACA regulatory sequences likely reflect differential regulation (Costa et al., 2023). To date, only one ECA regulatory protein (MIZ1) has been identified (Yamazaki et al., 2012).

ATPases show diverse patterns of localization that are both cell-type and species-dependent (Costa et al., 2023). For example, *Arabidopsis* possesses 4 Type IIA ECAs and 10 Type IIB ACAs, 5 of which are located at the PM. ACA9 is expressed in pollen tubes while ACA8 and ACA10 are expressed in vegetative cells. ACA12 and ACA13 13 are preferentially expressed under biotic or abiotic stress. Both type IIA ECA and IIB ACA are found in the ER membrane. Two type II ACAs (ACA4 and ACA11) are localized to the vacuole. ATPases and other transporters potentially serve two roles. Firstly, to restore and maintain resting $[Ca^{2+}_{cyt}]$ and secondly to charge up $Ca^{2+}$ stores involved in $Ca^{2+}$ release during signalling.

### 2.3.1. Plasma membrane transporters.

The roles of different $Ca^{2+}$-ATPases have been inferred from both genetic and inhibitor studies (Costa et al., 2023; Demidchick et al., 2018). Mutant studies have revealed significant redundancy and have indicated important roles for localization in determining function (Costa et al., 2023; Resentini et al., 2021a; Resentini et al., 2021b). Examples include impaired pathogen defence responses and attenuated $Ca^{2+}_{cyt}$ signals of PM-localized *Ataca8/10* double knockout mutants in response to flg22 (Frei dit Frey et al., 2012). Behera et al. (2018) showed that resting $Ca^{2+}_{cyt}$ was unchanged in *aca8/10* double mutants, which also showed decreased $Ca^{2+}_{cyt}$ signal amplitude and delayed recovery in response to external ATP compared with wild-type plants. This response was considered to reflect a degree of acclimation via modified expression of other transporters. By monitoring both $Ca^{2+}_{cyt}$ and pH these workers also demonstrated that $Ca^{2+}$ and pH fluxes were tightly linked.

### 2.3.2. Tonoplast transporters.

In contrast to mutants with disabled PM ATPases *Ataca4/11* double knockout mutants of vacuolar $Ca^{2+}$-ATPases have elevated baseline $[Ca^{2+}_{cyt}]$, elevated $Ca^{2+}_{cyt}$ response to $CO_2$ and enhanced defence responses (Hilleary et al., 2020). By imaging $[Ca^{2+}_{cyt}]$ with a YC-Nano65 sensor at both whole organ and sub-cellular scales, the flg22 response was found to be homogeneous across cells. Mis-localization of PM ACA8 suppressed the *aca4/11* phenotype, despite not having the same regulatory elements as ACA4/11. Perhaps surprisingly, while *aca4/11* mutants showed a significantly higher $[Ca^{2+}_{cyt}]$ signal in response to flg22, $[Ca^{2+}_{cyt}]$ returned to basal levels in a similar time frame as wild-type plants, suggesting the involvement of other efflux systems in re-establishing baseline $[Ca^{2+}_{cyt}]$. In contrast, knockout of the *Physcomitrium* tonoplast $Ca^{2+}$-ATPase *PCA1* gave higher $Ca^{2+}_{cyt}$ transients in response to high NaCl, which were of longer duration than wild type (Qudeimat et al., 2008). These studies also indicated that vacuolar $Ca^{2+}$-ATPases act very quickly to modulate the amplitude of the $Ca^{2+}_{cyt}$ signal.

CAX transporters belong to the multigene family of cation/$H^+$ exchangers (Demidchick et al., 2018, Pittman & Hirschi, 2016).

Plant CAX transporters have a lower affinity for $Ca^{2+}$ than $Ca^{2+}$-ATPases and transport $H^+$ and $Ca^{2+}$ in a 3:1 ratio (Demidchick et al., 2018; Dindas et al., 2021). *Arabidopsis* possesses 6 *CAX* genes (*AtCAX1–6*) and 5 further $Ca^{2+}$/cation antiporters that behave as $K^+$-dependent $Na^+$/$Ca^{2+}$ exchangers (Manohar et al., 2011; Maser et al., 2001, Shigaki et al., 2006). CAX transporters possess an N-terminal autoinhibitory domain and transport specificity is controlled by a 9 amino acid region between TM1 and TM2. Activity depends on $\Delta\mu H^+$ across the tonoplast, the degree of phosphorylation and interaction of regulatory proteins with the N-terminal region (Demidchick et al., 2018; Matthew et al., 2024; Pittman & Hirchi, 2016; Wang et al., 2024).

Despite their lower affinity, physiological and molecular studies indicate that CAX transporters play a pivotal role in the regulation of $Ca^{2+}_{cyt}$ dynamics. They also play important roles in the regulation of cytosolic and apoplast pH (Cho et al., 2012). *Arabidopsis det3* V-type $H^+$-ATPase mutants (Allen et al., 2000) had altered $Ca^{2+}$ dynamics in response to increased apoplastic $Ca^{2+}$ and ROS, displaying sustained stomatal guard cell $Ca^{2+}_{cyt}$ elevations rather than oscillations normally associated with these stimuli, likely reflecting defective $Ca^{2+}/H^+$ regulation *via* $Ca^{2+}/H^+$ transporters. However, *det3* mutants did show a normal pattern of $Ca^{2+}_{cyt}$ oscillations in response to ABA. Electrophysiological manipulation of tonoplast Vm combined with $Ca^{2+}_{cyt}$ imaging in *Arabidopsis* root hairs has provided further evidence for tonoplast $Ca^{2+}/H^+$ exchange in the regulation of $[Ca^{2+}_{cyt}]$ (Dindas et al., 2021). Depolarizing the tonoplast (i.e. rendering the cytosolic side more positive) elevated $[Ca^{2+}_{cyt}]$ and reduced $[H^+]_{cyt}$. This can be interpreted as reduced $\Delta\mu H^+$ leading to reduced $Ca^{2+}$ uptake into the vacuole in exchange for $H^+$. Hyperpolarizing the tonoplast produced the opposite effect. Two recent reports provide further direct evidence for the roles of CAX in the maintenance of $Ca^{2+}_{cyt}$ homeostasis. Bakshi et al. (2023) showed transcripts of *CAX2* and *ACA1* were rapidly upregulated in *Arabidopsis* plants subject to flooding or hypoxia. Moreover, *cax2* knockout mutants showed larger and more sustained $Ca^{2+}_{cyt}$ signals and enhanced survival in response to flooding. In a separate study (Conn et al., 2011) *Arabidopsis cax1/3* double mutants had reduced overall mesophyll $Ca^{2+}$ contents and apolastic free $Ca^{2+}$ that was 3-fold higher than wild-type plants. This was indicative of compensatory increased PM ATPase activity in response to reduced tonoplast $Ca^{2+}$ transport.

### 2.3.3. ER transporters.

There are numerous reports of disruption of ER $Ca^{2+}$-ATPases leading to altered $Ca^{2+}_{cyt}$ signalling and downstream responses (Costa et al., 2023). Analysis of $Ca^{2+}_{ER}$ dynamics in pollen tubes expressing ER-localized yellow cameleon 3.6 $Ca^{2+}$ sensor showed that CPA triggered growth arrest and a decrease in $[Ca^{2+}_{ER}]$ (Iwano et al., 2009). CPA also reduced the tip-focused $Ca^{2+}_{cyt}$ oscillations in the growing pollen tube tip and caused $[Ca^{2+}_{cyt}]$ to elevate in sub-tip regions indicating a key role for ER $Ca^{2+}$-ATPase in regulating the tip-focused $[Ca^{2+}_{cyt}]$ gradient. More specifically, *Arabidopsis* triple mutants *aca1/2/7* of ER-localized $Ca^{2+}$-ATPase show higher $Ca^{2+}_{cyt}$ response to flg22 or blue light, higher resting $[Ca^{2+}_{cyt}]$ and associated changes in downstream responses (Ishka et al., 2021). Triple *aca1/2/7* mutants also had slower recovery of $[Ca^{2+}_{cyt}]$ to resting levels following repeated cycles of elevated $CO_2$ as well as altered stomatal conductance (Jezek et al., 2021). Increases in $[Ca^{2+}_{cyt}]$ were also progressively reduced in mutants in response to successive $CO_2$ cycles, indicating that ACA-mediated recovery of the ER $Ca^{2+}$ store was required for response to repeated stimuli. A further role for $Ca^{2+}$-ATPase

in shaping $Ca^{2+}$ signatures comes from studies of nod factor signalling in *Medicago*. Silencing the nuclear envelope-localized $Ca^{2+}$-ATPase MCA8 blocked nod factor-induced nuclear $Ca^{2+}$ oscillations (Capoen et al., 2011).

A role for an ER-localised cation/$Ca^{2+}$ exchanger (CCX) in the regulation of both $[Ca^{2+}_{ER}]$ and $[Ca^{2+}_{cyt}]$ in *Arabidopsis* has also been demonstrated (Corso et al., 2018). Surprisingly, knockout of *AtCCX* resulted in decreased $[Ca^{2+}_{cyt}]$ and increased $[Ca^{2+}_{ER}]$ under salt and osmotic stress conditions. The underlying mechanism and role in cytosol-ER $Ca^{2+}$ exchange have yet to be fully elucidated.

## 3. Mitochondrial and chloroplast $Ca^{2+}$ transport

In animal cells, mitochondrial $Ca^{2+}$ uptake is critical for control of energy metabolism and mitochondria play a fundamental role in shaping spatio-temporal $Ca^{2+}_{cyt}$ increases. This occurs primarily via $InsP_3$-induced release of $Ca^{2+}$ from ER stores during $Ca^{2+}_{cyt}$ wave propagation and reuptake of $Ca^{2+}$ by mitochondrial $Ca^{2+}$ uniporters (MCUs) at specific locations where mitochondria make close contact with the ER (ER-mitochondrial contacts, ERMCs) (Katona et al., 2022; Lee et al., 2018). Mitochondria maintain a large inside-negative Vm across the inner mitochondrial membrane (Zorova et al., 2018) resulting in an inwardly-directed $\Delta\mu Ca^{2+}$, given mitochondrial matrix $Ca^{2+}$ $[Ca^{2+}_{mit}]$ of 100–200 nM (Finkel et al., 2015) (Figure 2). While there is little or no evidence for functional ERMCs in plant cells, recent work has shown that MCU proteins play a key role in $Ca^{2+}$ uptake. *Arabidopsis* possesses 6 MCU homologues (Teardo et al., 2017) and Ruberti et al. (2022) demonstrated in vitro $Ca^{2+}$ transport activity by *Arabidopsis* MCU and defective mitochondrial $Ca^{2+}$ uptake in a *mcu1/2/3* triple knockout mutant. However, this study also showed that $Ca^{2+}_{cyt}$ dynamics were unaffected in the triple mutant, indicating that mitochondria played a minimal role in $Ca^{2+}_{cyt}$ homeostasis. A homologue of the animal regulatory MCU-associated MICU proteins has been implicated in the regulation of MCU-mediated $Ca^{2+}$ uptake (Wagner et al., 2015). *Arabidopsis mice* mutants showed higher and faster mitochondrial ($Ca^{2+}_{mit}$) elevations in response to auxin and ATP while $Ca^{2+}_{cyt}$ remained unchanged. Thus, similar to animal cells, by modulating plant mitochondrial $Ca^{2+}$ uptake MICU shapes mitochondrial $Ca^{2+}$ signatures and helps to maintain mitochondrial $Ca^{2+}$ homeostasis.

Similar to mitochondria, chloroplasts have a large inside negative membrane potential (Svabo & Spetea, 2017) and a stromal $[Ca^{2+}]$ ($[Ca^{2+}_{strom}]$) of 50–200 nM (Frank et al., 2019; Hochmal et al., 2015), maintained by active transporters such as the thylakoid membrane-localized $Ca^{2+}/H^+$ exchanger CCHA1 (Wang et al., 2016). A chloroplast-localized homologue of mitochondrial MCU transporters (cMCU) has also been identified and shown to mediate chloroplast $Ca^{2+}$ uptake (Teardo et al., 2019). By using targeted aequorin reporters they showed that cMCU was required for stress-specific $Ca^{2+}_{strom}$ signatures. The dynamic nature of chloroplast $Ca^{2+}$ signalling is further illustrated by responses of $Ca^{2+}_{strom}$ to high light and temperature. By targeting YC3.6 to the cytosol, chloroplast and mitochondria in the green alga *Chlamydomonas*, Pivato et al. (2023) monitored elevated $Ca^{2+}$ specific to the chloroplast, which correlated with $H_2O_2$ production and was dependent on functional cryptochrome. Similarly, Flori et al. (2024) demonstrated sustained $Ca^{2+}_{strom}$ elevations in response to high light or $H_2O_2$ treatments that were independent of changes in $[Ca^{2+}_{cyt}]$ and accompanied chloroplast $H_2O_2$ accumulation in high light.

A further class of chloroplast $Ca^{2+}$ transporter, BICAT proteins, are involved in the elevation of $Ca^{2+}_{strom}$ on transfer from light to dark (Frank et al., 2019). Knockout mutations of *BICAT1*, which is located on the chloroplast envelope reduced the dark-induced $Ca^{2+}_{strom}$ signal, monitored with chloroplast-targeted aequorin. In contrast, knockout mutation of *BICAT2*, which transports $Ca^{2+}$ into the thylakoid lumen increased the light-dark $Ca^{2+}_{strom}$ signal and produced severe defects in chloroplast morphology.

## 4. Insights from modelling studies

A number of modelling studies have provided insights into the interactions between signalling and homeostatic components of the plant $Ca^{2+}$ signalling machinery. Bose et al. (2011) used a 4-component model, comprising 2 $Ca^{2+}$-permeable channels on the PM and endomembrane, together with 2 efflux systems – a PM $Ca^{2+}$-ATPase and an endomembrane $Ca^{2+}/H^+$ exchanger. They also factored in the ROS sensitivity of the endomembrane $Ca^{2+}$ channel and the buffering capacity of the cytosol. The model predicted that specific $Ca^{2+}$ signatures could be achieved by modifying the activities of ATPase and $Ca^{2+}/H^+$ exchangers. Dindas et al. (2021) combined electrophysiological manipulation of vacuolar Vm, $Ca^{2+}_{cyt}$ monitoring and modelling to demonstrate the role of a voltage-dependent vacuolar $Ca^{2+}$ homeostat involving tonoplast $Ca^{2+}/H^+$ exchange and vacuolar electrical excitability providing a clear demonstration that $Ca^{2+}$ fluxes across the tonoplast are important in regulating $Ca^{2+}_{cyt}$.

The "*On guard*" model (Jezek et al., 2021) presents arguably the most comprehensive analysis of transport and essential metabolism in predicting stomatal signalling patterns and behaviour. The model considers $Ca^{2+}$- and $H^+$-ATPases, along with cation and anion channel activities shown to be associated with stomatal responses to ABA or $CO_2$. The model accurately simulated elevated $[Ca^{2+}_{cyt}]$ and oscillations resulting from cyclic $Ca^{2+}$ influx across the PM, promoting much larger $Ca^{2+}$-induced $Ca^{2+}$ release (CICR) from endomembrane stores. Modelling also predicted, and experiments verified, a delay in $Ca^{2+}$ cycling that was enhanced in ER and tonoplast $Ca^{2+}$-ATPase mutants, identifying both endomembrane $Ca^{2+}$-ATPases and $Ca^{2+}$ channels as important targets for the stomatal closure response to high $CO_2$.

## 5. Outlook

There has been substantial progress in understanding the essential components of the plant $Ca^{2+}$ homeostatic machinery and how these contribute to the shaping of $Ca^{2+}$ signals in response to a wide range of stimuli in different plant, tissue and cell types. This progress has been facilitated largely by advances in molecular characterization of key transporters, along with the development of targeted genetically encoded indicators with differing affinities suited to imaging $Ca^{2+}$ in the cytosol and other cellular compartments, along with increasingly refined modelling simulations. Technological advances in microscopy from sub-cellular to whole plant imaging have further enabled spatio-temporal $Ca^{2+}$ signalling patterns to be analysed in far greater depth than previously possible. Discoveries enabled through studies of a wider range of organisms, such as the recently described MID1-COMPLEMENTING ACTIVITY (MCA) PM $Ca^{2+}$-permeable channel with a role in the regulation of cell proliferation in the liverwort *Marchantia polymorpha* (Iwano et al., 2025) will provide further insights into the roles and evolution and of plant $Ca^{2+}$ signalling components.

Despite these advances, there is still a need to further define the roles of endomembrane stores as sources and/or sinks for $Ca^{2+}$. There has been substantial progress in understanding how both PM and endomembrane $Ca^{2+}$-ATPases and CAX transporters can shape different $Ca^{2+}$ signatures, though with still much to discover. While CNGCs have been identified as the major $Ca^{2+}$ release pathway from the nuclear envelope underlying nuclear $Ca^{2+}$ oscillations during symbiotic signalling, our more general understanding of the roles of endomembrane $Ca^{2+}$ release mechanisms is less advanced. How is an initial $Ca^{2+}$ influx across the PM augmented and amplified by the release of $Ca^{2+}$ from internal stores? Of particular importance is the ongoing need to identify ER $Ca^{2+}$ release mechanisms in the absence of molecular homologues of $InsP_3$ receptors that are widespread in animal cells and play a central role in $Ca^{2+}$ signalling. There is also a need to further understand the interactions between different endomembrane compartments. In animals, interactions between ER and mitochondria are pivotal in defining spatiotemporal $Ca^{2+}$ signalling. To date, no similar evidence exists for plant cells. Are there similar direct interactions between the ER and mitochondria or chloroplasts in plants? The vacuole represents the largest potential $Ca^{2+}$ stored in most plant cells. However, while tonoplast $Ca^{2+}$ transporters have been shown to play a role in $Ca^{2+}_{cyt}$ homeostasis, apart from the recently discovered tonoplast PIEZO channels (Mousavi et al., 2021; Radin et al., 2021) there is no fully characterized channel-mediated mechanism for vacuolar $Ca^{2+}$ release associated with specific $Ca^{2+}$ signalling events.

Finally, in animals, store-operated $Ca^{2+}$ entry (SOCE) is an important mechanism for recharging ER stores during repetitive $Ca^{2+}$ signalling and is a key component in shaping $Ca^{2+}$ signatures. However, the molecular machinery for SOCE – STIM proteins that sense $[Ca^{2+}_{ER}]$ and ORAI channels in the PM that allow $Ca^{2+}$ entry in close proximity to ER-PM contact points (Lunz et al., 2019; Wang et al., 2014) are absent in embryophytes, though ORAI proteins are present in the green lineage as far as gymnosperms (Edel et al., 2017). This begs the question of whether SOCE exists, at least in embryophytes.

## Acknowledgements

C.B. is grateful to the Marine Biological Association for its continued support in the preparation of this manuscript.

**Competing interest.** The authors declare none.

**Data availability statement.** This review article does not rely on original data or resources.

**Author contributions.** C.B. conceived the study. C.B. and G.L.W. wrote the article.

**Funding statement.** This work received no specific grant from any funding agency, commercial or not-for-profit sectors.

**Open peer review.** To view the open peer review materials for this article, please visit http://doi.org/10.1017/qpb.2025.2.

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
