## [Reviewer Report]

This is a timely review in a field that has seen an steady increase of new data and contentious models in the past couple of decades. The section “cytosolic buffers” makes full justice to the journal title, and demonstrate a global view of the Ca2+ buffering system from a quantitative angle that only a few can write with such authority. And denotes the urgent need to come back to more quantitative methods, namely electrophysiology, that are slowly eroding to the most popular and easy to approach genetic and imaging methods. This section should be a must to anyone wanting or needing to do research in this area. The remaining is a competent and well organized “mise au point” of the continuously growing repertoire and functions of known genes for Ca2+ channels transporters and pumps, and a positive addition as an update to other reviews that cover the field of Ca2+ buffering and regulation. As the aa. correctly point, plants still lack an integrating model of store operated Ca2+ buffering operated by small diffusing ligands, and this gap is slowing down considerably advances in the understanding of many spatial and temporal aspects of Ca2+ signaling. The text is in very good shape, some notes are added as minor additions.

1- The 1st section (Cytosolic buffers) could do better with more specific references on the different quantitative aspects and how these numbers were obtained. The entire section comes only with a hand-full of references, when it would be important to reference the original literature from where these number and mechanisms were deduced from. Even if covered on previous reviews, it is important to check methods and controls and how these quantities were estimated, special for younger researchers that had not access to what doing research in this area used to be about. The part starting in “Clearly, if all...” to “ ...Ca2+ buffering and or removal” is a good example of a noted absence of references.

2- Reference “Demidchick et al.” lakes a date.

3- Define what “DELTAµCa2+” means.

4- page 6 last line of 1st paragraph “defined” is repeated.

5- The finding of a new channel contributing to Ca2+ regulation was just described in Marchantia and could be added as an e.g. of other models ( Plant Physiology, kiae613, https://doi.org/10.1093/plphys/kiae613). In that respect it should be noted that most of the references to cellular models are focused on the stomata, leaving out of this review important accounts in other systems (e.g. pollen tubes,or Physcomitrium protonema.)

---

## [Reviewer Report]

Dear Editor,

This is an excellent review on the calcium transport systems in plants and the mechanisms that provoke cytosolic Ca2+ signals. The manuscript starts with some calculations that highlight the importance of the Ca2+-buffering capacity of cells. It continues with a discussion of channels that are involved in the release of Ca2+ into the cytosol and the free Ca2+ concentrations of subcellular compartments. This part is followed by the description of ion-transporters remove Ca2+ from the cytosol and store into the apoplast and various organelles. The review ends with a discussion of modelling approaches and gives an outlook that addresses open questions on Ca2+ homeostasis in plants cells.

Overall, the manuscript gives a good overview of the literature, without going into too much detail. I only have some points that the authors may consider, while preparing the final version of their manuscript.

1. In 2021 two groups showed that cytosolic immune receptors can form Ca2+ permeable channels in the plasma membrane; Jacob et al., Science (DOI: 10.1126/science.abg7917) and Bi et al., Cell (DOI 10.1016/j.cell.2021.05.003).

It is worthwhile to mention these papers.

2. Page 3. It may be good to point out that genes encoding voltage-dependent Ca2+ channels in the plasma membrane, only were found in animals, so far. The gene encoding the predicted hyperpolarization activated channel of guard cells has not been identified yet.

3. Please consider the following with respect to the role of TPC1 in Ca2+ signaling.

- Peiter et al. (2005) found that tpc1 stomata fail to close in response to extracellular Ca2+, while Islam et al. (2010) only observed a reduced Ca2+-induced stomatal closure of tpc1 stomata, compared to wild type.

- The impact of TPC1 on long distance Ca2+ signals was also shown by Kiep et al. (2015), New Phytologist (DOI10.1111/nph.13493).

4. Are the authors sure that CAX transporters are translocating Ca2+ and H+ at a ratio of 1:1? Dindas et al. (2021) concluded that the Ca2+/H+ ratio should be at least 1:3.

5. Note that the following interesting papers on the role, and regulationm of CAX-transporters were published: Wang et al. 2024 (DOI 10.1038/s41586-024-07100-0), Mathew et al. 2024 Plant Cell Environment (DOI 10.1111/pce.14756).

The following mistakes were found.

1. Introduction, page 1, last sentence. “(CDPKS)” should be “(CDPKs)”

2. Introduction, page 1, last sentence. “Edel et al., 2017 is cited twice.

3. Page 2, “(Thorr et al. 2019)” should be (Thor et al., 2019).

4. Page 4, line 6. “direted” should be “directed”.

5. Page 6, The reference “(Hilleary et al. 2024)” probably should be “(Hilleary et al. 2020)”.

6. Page 7, the second last sentence is confusing. It is likely that “the absence of” ACA pumps caused a latency in response.

7. The reference list needs a check, the following mistakes were found:

- Brownlee and Wheeler (2023) are listed twice

- In Dindas et al. (2021) “Roelfesma, R.G.” should be “Roelfsema, M.R.G.”

- In Frei dit Frey et al. (2012) „H ¨ aweker, H.” should be „Häweker, H“.

---

## [Editor Report]

Desr authors,

The manuscript has been seen by two experts. Both reviewers are very positive and have only a few suggestions. Please consider those in a minor revision. Thank You for your valuable contribution to the the Research Topic “Quantitative approaches to cellular aspects of plant ion homeostasis”.

Best regards,

Ingo

---

## [Editor Report]

Dear Colin and Glen,

thank you very much for the careful revision of your manuscript. All pending issues have been resolved. Thanks again for your very valuable contribution. It is highly appreciated.

Regards, Ingo